## [Decision Letter · Decision Letter 0]

5 Dec 2021

PONE-D-21-30685Spanish validation of the internal corporate social responsibility questionnairePLOS ONE

Dear Dr. Bayona,

Thank you for submitting your manuscript to PLOS ONE. After careful consideration, we feel that it has merit but does not fully meet PLOS ONE’s publication criteria as it currently stands. Therefore, we invite you to submit a revised version of the manuscript that addresses the points raised during the review process.

Please note that both reviewers point out very relevant concerns about your work, which should be addressed satisfactorily for the paper to be publishable. I think you could undertake the task, even if the effort involved is high and the success chances low, hence I invite you to resubmit an improved version of your paper for reconsideration if  you are convinced that you have overcome the shortcomings of the present version.

We look forward to receiving your revised manuscript.

Kind regards,

Iván Barreda-Tarrazona, PhD

Academic Editor

PLOS ONE

Journal Requirements:

Reviewers' comments:

Reviewer's Responses to Questions

**Comments to the Author**

1. Is the manuscript technically sound, and do the data support the conclusions?

Reviewer #1: Partly

Reviewer #2: No

2. Has the statistical analysis been performed appropriately and rigorously? 

Reviewer #1: Yes

Reviewer #2: I Don't Know

3. Have the authors made all data underlying the findings in their manuscript fully available?

Reviewer #1: Yes

Reviewer #2: Yes

4. Is the manuscript presented in an intelligible fashion and written in standard English?

Reviewer #1: Yes

Reviewer #2: Yes

5. Review Comments to the Author

Reviewer #1: - In page 3 the authors should include the header INTRODUCTION

- In page 4 the authors explain the objective of the research like this: "to offer a valid and reliable Spanish adaptation of the ICSR Q developed by Mory using a sample of workers from Bogotá, Colombia". This claim is confusing, they are refering to the language, not to the country. Maybe they should use the word "Spanish-speaking" (and throughout the whole paper).

- In page 6 they refer to the Mory´s questionnaire tested in Germany. It would be advisable to explain the sample characteristics. In this page there is also a mistake: workface (maybe workforce?).

- Hypothesis 1 needs more theoretical support.

- In page 7 authors should explain and theorethically support the variables affective organizational commitment and normative organizational commitment in order to support hypotheses 2a and 2b.

- In page 10 they say: both English and Spanish versions are shown in the Appendix. I have only seen the Spanish version.

- In page 11 there is another mistake: firs (maybe first?).

- To test the second set of hypotheses they examine the correlations. I think it is a very poor analysis, there are much more appropriate test to carry out this.

- Page 18, limitations. They say: "the proportion of informal work is lower" (informal? is it right?)

Reviewer #2: Thank you for the invitation to review this manuscript, and to the authors for addressing this relevant topic regarding Internal Corporate Social Responsibility. Unfortunately, however, the paper is very poorly structured and its main contribution is unclear. I think that the paper has the following weaknesses:

-Abstract/Introduction.

No clear introduction or research framework that provides a foundation for a research problem. In fact, there is not any section entitled as “Introduction”.

The purpose of this study is “to validate a Spanish version of the internal corporate social responsibility questionnaire”. This purpose is weak. First, it is crucial to define a tangible research question or problem. Readers need a strong reason why this study is necessary, what is the contribution after considering the paper of Mory et al. (2016)? Why is necessary a Spanish validation of the Mory et al. (2016) survey?

-Internal Corporate Social Responsibility section. The paper mentions ISO 26000 guidelines for social responsibility and KLD indexes, however, it is necessary to explore in depth the content of this standards regarding the labor aspects. Other fundamental standards regarding occupational health and safety such as OHSAS 18001 have been not considered.

In this part, the authors mentioned the social exchange theory as a framework for the theoretical part of the paper. However, the presented hypotheses need a strong connection with this framework. In particular, the Hypothesis 3 requires a further argumentation and it is important to explain the justification of this hypothesis, since it seems that by definition, permanent contracts present higher employee stability and consequently higher evaluations of ICSR.

-Method.

Misunderstanding with the information included. In the measures, it is mention that both English and Spanish versions of questionnaires are included and only the Spanish version is presented. The internal corporate social responsibility is measured by 48 items, however in the Spanish version of questionnaire only 45 questions are included. This requires a further explanation.

Could the data collection bias the results? Almost 50% of the sample comes from a university and the rest from private participants in a human resource management course.

Much more description is needed to improve confidence in the data being used, for instance, when the data was accessed?

-Results

The comments in this section are very superficial. And additional explanation and argumentation of the results is needed.

-Conclusion

The conclusion section is very poor.

6. PLOS authors have the option to publish the peer review history of their article (what does this mean?). If published, this will include your full peer review and any attached files.

Reviewer #1: No

Reviewer #2: No

---

## [Author Response · Author response to Decision Letter 0]

2 Feb 2022

All modifications are highlighted in the document 

Response to reviewer # 1

Point # 1: In the page 3 the authors should include the header Introduction.

Response 1.

A. Thank you for your comment, the “introduction” section has been included on page 3.

Point # 2: In page 4 the authors explain the objective of the research like this: "to offer a valid and reliable Spanish adaptation of the ICSR Q developed by Mory using a sample of workers from Bogotá, Colombia". This claim is confusing, they are refering to the language, not to the country. Maybe they should use the word "Spanish-speaking" (and throughout the whole paper).

A. Thank you for your observation, The word "Spanish - speaking" has been adjusted throughout the document to refer to the language in which the instrument has been validated (ICSR).

Point # 3: In page 6 they refer to the Mory´s questionnaire tested in Germany. It would be advisable to explain the sample characteristics. In this page there is also a mistake: workface (maybe workforce?)

A. Thank you for your comment, additional information about the sample was included (pp. 9-10):

“The original questionnaire was developed using a sample of 2081 employees from an international operating pharmaceutical company with its headquarters located in Germany, the sample distribution was 50.6 male, 17.9% were under 29 years, 24.8 from 30-39 years, 35.2% from 40-49 years, and 22% were older than 50 years. The tenure of the sample was up to 2 years 17.3%, from 2-5 years 16.7%, from 6-10 years 22%, and 10 years and more 44% (7).”

B. The typo was corrected (now in page 9)

Point # 4: hypothesis 1 needs more theorical support.

A. Thank you for your observation, regarding to hypothesis 1: The Spanish – speaking version of the ICSR Q retains a seven factor structure. We have included addition support, included a description of ISO 26001, KLD, and OHSHA, the description of the original sample, and information about the inclusion of Colombia in the OECD, and how the HRM practices have changed during the last decades. now this section reads as follows (pp 6-11):

CSR have evolved from their beginning as philanthropic activities to an structured area of management that deals with the impact of organizations in their context. Within the evolution of this area of management, the responsibilities with external stakeholders were the first to become accepted the communities that could be impacted by the operations of the company. Later, the responsibilities of organizations were extended to their employees, primary to their contractual and compensation duties; this inclusion is present in major models of CSR as for example the Global Compact, which includes four labour principles related to the respect of association, elimination of forced labor, abolition of child labor and discrimination (9), CSR certifications as in ISO 26000, which includes a “core subject” of labour practices (10), or in burse indexes as KLD, which include a factor called employees and supply chain (11).

ISO 26001 considers an organization's labor practices, including all work-related policies and practices of the organization, such as labor relations, working conditions and social protection, social dialogue, health and safety development of workers and people. ISO 26001 focuses on 5 topics in the framework of labor practices, which are: labor and labor relations, working conditions and social protection, social dialogue, occupational health and safety, and human development and workplace training (12). In addition to the above, it recommends that organizations provide skills development, training and learning, and career advancement opportunities to all workers at all stages of their work experience in a fair and non-discriminatory manner. 

Social rating agencies are responsible for determining whether a company complies with the principles of social responsibility, i.e., they perform a social audit, and then the agency is responsible for certifying a company when it complies with all the requirements of responsible practice. Another task of these agencies is to collect, analyze and organize information on the social behavior of companies to provide investors with homogeneous and comparable information. The MSCI ESG STATS index, known as KLD Research & Analytics Inc, ranks 3,000 companies located in the United States. One dimension considers social aspects and links community investment, diversity and equal employment opportunity, human rights and labor relations. They are also responsible for rating investments by companies seeking to promote women or minorities to senior executive positions. They also highlighted investments by companies working on labor rights initiatives in the U.S. and ultimately focused on understanding employee concerns and retirement benefits (13).

Likewise, the overall objective for the creation of the OHSAS 18001 standard was "to support and promote good practices in the field, maintaining a balance with the socioeconomic needs of the organization", and as a general objective to improve the working environment. in terms of health and safety, which in turn has greatly helped to help prevent accidents. OHSAS 18001 has become an important standard, accepted by different business sectors and companies of all sizes, and certified organizations are highly competitive companies (14). Items considered in OHSAS 18001 include: understanding workers' needs and expectations, leadership and worker participation, and risk assessment in occupational health and safety systems. In addition, the standard covers all aspects of the physical environment consisting of physical, biological and, most importantly, ergonomic factors that can present risks to the health and safety of workers and is generally designed to ensure personal health. opportunities and flexibility that companies provide to their employees to monitor and promote their own health (15).

It is important to note that, although included in standards, certifications and indexes, the “workers dimension” is generally understood from a legal perspective (i.e., if the organization fully respect the local labour normative), leaving aside the stakeholder expectations component of the CSR definition, as not all expectations are tied to the legal duties of organizations. This is also true from the standpoint of organizational reputation, as there is growing number of consulting firms and databases that rank organizations based on their quality of human resources practices (e.g., benefits plan, engagement practices). However, although popular among managers, these evaluations are by no means congruent to any conceptual framework, as their objective is related to consulting and professional advice. Some theoretical analyses have been developed based on the consequences of CSR on individuals, as organizational commitment, job satisfaction, OCB or job performance (2), but these outcomes cannot be considered responsibilities per se.

A theoretical framework that can include these responsibilities is the social exchange theory, which is based in reciprocity, if one party voluntary provides a benefit to another, invoking an obligation to reciprocate by providing some benefit in return (16). These exchanges between individuals goes beyond the monetary level; these social relations are characterized by a continuous exchange process in which the individual trade resources (e.g. their knowledge) while striving for a balance in the give-and-take of these process (17). In social exchange theory, there are six different resources that individuals can exchange: love, services, goods, money, information, and status and could be present in any social relation (18). 

“The rule of reciprocity applies in the case of CSR, because it implies voluntary actions by the firm to support well-being of employees […]. Therefore, employees may feel obliged to reciprocate these voluntary investments” (11: p.567). Following this logic, Mory et al. (7) translated this framework to the organizational context, in which, love is the care that organizations dedicate to employees (including job stability); Services are company activities directed to the development of employee skills, as training and education, or a healthy work-life balance. Goods are products and objects creating an appropriate physical and psychological work environment, as well as preventing discrimination against groups and minorities. Money consists of symbolic gifts, involvement with financial resources, and sharing of profits with employees in a voluntary way. Information is education, while status is prestige and recognition provided by the organization resulting in empowerment (7).

Based on this classification of resources from the social exchange theory, Mory et al. (7) developed the ICSR Q built on seven dimensions: “...the resource ‘love’ [is related] to companies’ assurance of employment stability, ‘services’ to the promotion of employees’ skills development, workforce diversity and work-life balance, ‘goods’ to a working environment which ensures health and safety at work, ‘money’ to companies’ tangible employee involvement through shares as well as ‘information’ and ‘status’ to employees’ empowerment in the decision-making process” (p. 5). This questionnaire was tested in Germany were the seven-factor structure was confirmed with adequate levels of validity and reliability. 

The original questionnaire was developed using a sample of 2081 employees from an international operating pharmaceutical company with its headquarters located in Germany, the sample distribution was 50.6 male, 17.9% were under 29 years, 24.8 from 30-39 years, 35.2% from 40-49 years, and 22% were older than 50 years. The tenure of the sample was up to 2 years 17.3%, from 2-5 years 16.7%, from 6-10 years 22%, and 10 years and more 44% (7).

Although Germany and Colombia have very different cultural profiles; for example, Colombia have higher power distance, but Germany is far more individualistic and long-term oriented (19), during the last 30 years, Colombia have opened its borders to international commerce, and modified its labor regulations, what has resulted in Colombian companies being more aligned with current global management practices. In addition, Colombia was accepted at the OECD in 2020 after a seven year process in which the country had to introduced major reforms to align its legislation, policies and practices to OECD standards (20), improving several aspects on employment and work, as the strength of the labor legal codes which give more guaranties to employees to improve their employment stability, working environment and workforce diversity. In relation with the organizations, the human resources management have been changing during the last 15 years, as skill development, diversity, work-life balance and empowerment are considered key factors in the current HRM in Colombia (21). Finally, in the last decade, human resource management consulting firms have entered in the local market, assessing the quality of human resource management practices and how “attractive” organizations are to potential employees; these consulting firms, also known as third party employment branding have become very popular in Colombia, and being listed in their rankings is very appreciated by organizations, for its reputational value, but also because these certifications are associated with lower turnover rates among employees (6). In particular, these consulting firms evaluate some practices that are common to ICSR, as skill development, work environment, diversity, work-life balance, empowerment and compensation. 

Because of this, it is reasonable to expect that a Spanish – speaking adaptation of the ICSR Q should preserve the original seven-factor structure (figure 1), leading to the first study hypothesis:

Hypothesis 1: The Spanish – speaking version of the ICSR Q retains a seven-factor structure.

Point # 5: In the page 7 authors should explain and theoretically support the variables affective organizational commitment and normative organizational commitment in order to support hypotheses 2a and 2b.

A. Thank you for your comment, a theoretical explanation to support affective and normative organizational commitment was included. Now this section reads as follows (pp. 11-13)

Workers are more attracted to organizations that promote ICSR practices and hence demonstrate greater interest in fairness, equality, and loyalty. “The organization provides benefits to its employees beyond its legal and financial obligations (voluntary) and employees feel obliged to pay back these voluntary investments. Consequently, it stimulates direct social exchange relationships between employees and their organization” (16, p. 567). There is evidence that higher levels of ICSR components are positively associated with higher organizational commitment. For example, Tarcan et al. (22) proposed that organizational commitment reacts positively to initiatives related to transformational leadership, job satisfaction and organizational trust. In addition, in a study of 191 hotel employees in Indonesia (23), a confirmatory factor analysis revealed that ICSR practices were associated with a decrease in employee turnover and increase in organizational commitment (β = 0.27). These findings suggest that ICSR practices may help employees to address challenges and opportunities from the external environment and facilitate the development of a highly talented, motivated, and committed workforce (24). Mory et al. (7) also reported a positive and significant relationship between ICSR and both affective and normative components of organizational commitment. 

The relation between CSR and organizational commitment have been explored, as “more socially responsible corporations are more attractive to potential employees and that they may therefore benefit from larger applicant pools […] and a more committed workforce because ‘employees will be proud to identify with work organizations that have favourable reputations’ (25, p. 1702). 

Farooq et al. (2014) reported that CSR toward employees (which includes actions as career opportunities, organizational justice, family-friendly policies, safety, job security, and union relations) is the strongest predictor of employees´ trust, identification, and affective organizational trust (16). 

 In relation to ICSR, Brammer et al. (2007) reported a positive relation between organizational justice and organizational commitment, in particular, these authors understand organizational justice as an special kind of ICSR, as “beneficial actions directed at employees create a reason for employees to reciprocate with their attitudes and their behaviours. At the same time a positive relationship may be expected between procedural justice and affective commitment because employees may be expected to identify with ethical organizations” (25, p. 1705). Following this argument, we can expect that, as all the dimensions considered in the ICSR Q are beneficial actions directed to employees, there could be a relation between these seven actions and organizational commitment.

Affective commitment refers to identification, involvement, and emotional attachment to an organization. Therefore, employees with strong Affective commitment stay with the organization because they want to. Normative commitment refers to commitments based on a sense of obligation to the organization (26).

Recent studies based on theory of social exchange establish a positive link between employee’s perceptions of their companies and CSR, developing CSR practices creates a good reputation among employees. Likewise the CSR toward employees the firm´s actions must ensure the well – being and support of its employees, including career opportunities, organizational justice, family – friendly policies, safety, job security, and union relations (16). 

A study conducted in different business sectors in Romania investigated the relationship between factors such as organizational commitment and social responsibility towards employees, and the survey response rate was 70%, showing a correlation of 0.447. The results showed that social responsibility to employees was positively correlated with organizational commitment and was the variable with the highest value, probably because the sample was composed of employees. This was one of the first articles to show the construction of a solid relationship between employees and employers (27).

We therefore expect ICSR to be positively and significantly related to affective and normative organizational commitment.

Point # 6: In the page 10 they say: both English and Spanish versions are shown in the Appendix. I have only seen the Spanish version.

A. Thank you for your observation, we have included both English and Spanish versions (pp. 33-37)

Point # 7: In the page 11 there is another mistake: firs (maybe first?).

A. Thank you for your comment, the spelling error on page 18 was corrected. 

Point # 8: To test the second set of the hypotheses they examine the correlations. I think it is a very poor analysis, there are much more appropriate test to carry out this. 

A. Thank you for your observation, we changed the method, and now we used the standardized values of the CFA. We have included the new results in page 21 and included an additional figure (number 3) in this same page. The new results about H2 now reads as follows:

Our second set of hypotheses involved exploring whether ICSR was related to both affective and normative organizational commitment. To test this, we examined the standardized values of the CFA between ICSR and organizational commitment. As we can see in figure 3, ICSR was significantly related to both affective organizational commitment (β= .81, p≤ .01) and normative organizational commitment (β= .80, p≤ .01). These results indicate full support for Hypothesis 2a and 2b.

Fig. 3. Structural model for ICSRQ and organizational commitment

Note. Standardized loadings. **p < .01.

Point # 9: They say: “the proportion of informal work is lower” (informal? Is it right?).

A. Thank for your observation, the paragraph was confusing, we have modified in order to gain clarity, now the paragraph reads as follows (p.26):

Despite our positive results, two limitations can be highlighted. The first is the sampling method. Although two different samples were obtained, they were collected in the urban context of Bogotá, where there is a greater representation of certain service organizations and the quality of job positions is higher; however, the proportion of the informal labor sector is around 50%. Although our conclusions characterize the sample of urban workers, future research should aim to represent other regions and economic sectors.

 Response to reviewer # 2

Point # 1: Abstract/Introduction

No clear introduction or research framework that provides a foundation for a research problem. In fact, there is not any section entitled as “Introduction”. The purpose of this study is “to validate a Spanish version of the internal corporate social responsibility questionnaire”. This purpose is weak. First, it is crucial to define a tangible research question or problem. Readers need a strong reason why this study is necessary, what is the contribution after considering the paper of Mory et al. (2016)? Why is necessary a Spanish validation of the Mory et al. (2016) survey?

A. Thank you for your comment, the “introduction” section was included in the document, we also have modified the introduction section, including more information about why is necessary to have an Spanish version of the ICSR Q. Finally, we have also modified the purpose. Now this section reads as follows (pp. 3-6)

Corporate social responsibility (CSR) is the “context-specific organizational actions and policies that take into account stakeholders’ expectations and the triple bottom line of economic, social, and environmental performance” (2: p.855). However, although CSR is a highly dynamic area of research and practice, it have been primarily studied from the institutional or organizational level, with far less emphasis on the individual level (2). In addition, the majority of CSR research at the individual level deals with the impact of CSR policies and practices on workers, with very limited research considering the workers themselves as subjects of responsibility.

This is somehow contradictory, as CSR includes in the social performance dimension, characteristics as fair labor practices and wages. However, due to the inclusion of decent work as a sustainable development goal (3), the rise of talent management as a new approach to improve attraction and retention of employees (4), job quality as a goal (5), and third party employment branding organizations as legitimate players in the building of reputation of organizations (6), organizations are more prone to offer workers certain practices that improve their quality of working life and goes beyond the minimum requirements stablished by labor law. 

These sets of practices that traditionally are attached to human resources, are also known as internal corporate social responsibility (ICSR) which is defined as “socially responsible behaviour by a company towards its employees. This behaviour is mainly expressed through employee-oriented CSR activities such as fostering employment stability, a positive working environment, skills development, diversity, work-life balance, empowerment and tangible employee involvement” (1: p.2). These practices of ICSR are not by any means new to organizations, as they are nowadays present as standard work practices following the trend of talent management and quality of work; however, what have changed is that these practices are now understood not as additional benefits, but as responsibilities that should be guarantee to all workers.

Several authors have revealed that actions aimed at internal social responsibility, considering employee training and equal opportunities, contribute to increase job satisfaction and, organizational commitment. To achieve this objective, this study traces internal social responsibility from the social exchange theory to provide theoretical background and explore the relationship between internal social responsibility and organizational commitment in companies in Bogota, Colombia. To our knowledge, no model has been proposed to explore the impact of employers on employees' internal perceptions of social responsibility. Therefore, a theoretical synthesis of the impact of intra-organizational social responsibility on affective and normative commitment is needed. Numerous surveys have revealed that internal social responsibility leads to better organizational performance, especially financial performance, but more recent studies, such as Mory et al. (2016) tried to find out the relationship with organizational commitment (8).

To date, few studies have empirically examined various aspects of the dimension of internal social responsibility and its impact on employees' organizational commitment, as existing research has employed rather basic empirical methods.. This study focuses on how employees perceive internal social responsibility from the individual as well as organizational perspective and, therefore, to explain employees' perceptions of social behavior imposed by the organization on employees. In addition, employees' organizational commitment depends on the organization's efforts to develop internal social responsibility practices, so it is based on social exchange theory and applies a structural model linking individuals' perceptions of internal social responsibility versus employees' emotional and normative commitments.

In addition, there is a lack of validated instruments that assess these components of responsibility from the standpoint of workers, one of the few available is the one developed by Mory et al. (7) which is based in the social exchange theory and assess seven factors of ICSR. To our knowledge, there are no available Spanish-speaking adaptations of this instrument, which will allow organizations to assess the full range of key ICSR dimensions. This kind of measure will help organizations to check the congruence between their own efforts to improve talent management and quality of working life, and the actual perception of these variables by their workforce. 

Although various instruments are available to measure different aspects of ICSR, to our knowledge, there is no other instrument as comprehensive as the ICSR Questionnaire (ICSR Q), which offer the possibility to evaluate seven different practices of ICSR, and their availability Spanish language is limited. Such scale would enhance our understanding of how internally motivated practices could improve organizational commitment and could give managers a snapshot of the current state of these practices in their organizations, offering actionable information about what their employees appreciate in their organizations.

The present study aimed to determine the psychometric properties of the ICSR Q in its Spanish – speaking version, in particular, our objectives are: 1) to present an Spanish version of ICSR Q, 2) to confirm the seven-dimension structure of the scale, 3) to determine the internal consistency of the ICSR Q, 4) to explore the relation between the ICSR Q and organizational commitment, and 5) to examine the association between ICSR dimensions and type of contract. 

The remainder of our article is structured as follows. The ensuing section presents the ICSR theory and hypotheses development, we then explain the methodology and present the empirical data used for testing the hypotheses. Finally, the results of the empirical analysis are discussed and their implications for both practice and research are presented.

Point # 2: Internal Corporate Social Responsibility section. The paper mentions ISO 26000 guidelines for social responsibility and KLD indexes, however, it is necessary to explore in depth the content of this standards regarding the labor aspects. Other fundamental standards regarding occupational health and safety such as OHSAS 18001 have been not considered.

A. Thank for your observation, we have included additional information about ISO, KLD, and OSHAS. Now this section reads as follows (pp.6-8): 

CSR have evolved from their beginning as philanthropic activities to an structured area of management that deals with the impact of organizations in their context. Within the evolution of this area of management, the responsibilities with external stakeholders were the first to become accepted the communities that could be impacted by the operations of the company. Later, the responsibilities of organizations were extended to their employees, primary to their contractual and compensation duties; this inclusion is present in major models of CSR as for example the Global Compact, which includes four labour principles related to the respect of association, elimination of forced labor, abolition of child labor and discrimination (9), CSR certifications as in ISO 26000, which includes a “core subject” of labour practices (10), or in burse indexes as KLD, which include a factor called employees and supply chain (11).

ISO 26001 considers an organization's labor practices, including all work-related policies and practices of the organization, such as labor relations, working conditions and social protection, social dialogue, health and safety development of workers and people. ISO 26001 focuses on 5 topics in the framework of labor practices, which are: labor and labor relations, working conditions and social protection, social dialogue, occupational health and safety, and human development and workplace training (12). In addition to the above, it recommends that organizations provide skills development, training and learning, and career advancement opportunities to all workers at all stages of their work experience in a fair and non-discriminatory manner. 

Social rating agencies are responsible for determining whether a company complies with the principles of social responsibility, i.e., they perform a social audit, and then the agency is responsible for certifying a company when it complies with all the requirements of responsible practice. Another task of these agencies is to collect, analyze and organize information on the social behavior of companies to provide investors with homogeneous and comparable information. The MSCI ESG STATS index, known as KLD Research & Analytics Inc, ranks 3,000 companies located in the United States. One dimension considers social aspects and links community investment, diversity and equal employment opportunity, human rights and labor relations. They are also responsible for rating investments by companies seeking to promote women or minorities to senior executive positions. They also highlighted investments by companies working on labor rights initiatives in the U.S. and ultimately focused on understanding employee concerns and retirement benefits (13).

Likewise, the overall objective for the creation of the OHSAS 18001 standard was "to support and promote good practices in the field, maintaining a balance with the socioeconomic needs of the organization", and as a general objective to improve the working environment. in terms of health and safety, which in turn has greatly helped to help prevent accidents. OHSAS 18001 has become an important standard, accepted by different business sectors and companies of all sizes, and certified organizations are highly competitive companies (14). Items considered in OHSAS 18001 include: understanding workers' needs and expectations, leadership and worker participation, and risk assessment in occupational health and safety systems. In addition, the standard covers all aspects of the physical environment consisting of physical, biological and, most importantly, ergonomic factors that can present risks to the health and safety of workers and is generally designed to ensure personal health. opportunities and flexibility that companies provide to their employees to monitor and promote their own health (15).

In this part, the authors mentioned the social exchange theory as a framework for the theoretical part of the paper. However, the presented hypotheses need a strong connection with this framework. In particular, the Hypothesis 3 requires a further argumentation, and it is important to explain the justification of this hypothesis, since it seems that, permanent contracts present higher employee stability and consequently higher evaluations of ICSR. 

B. Thank for your observation, we have included additional information about the connection between social exchange theory and our hypotheses. In relation to H3 we also included additional justification related to the usefulness of this hypothesis to test construct validity. Now this section reads as follows (pp. 11-15):

Additional evidence of ICSR Q validity

Workers are more attracted to organizations that promote ICSR practices and hence demonstrate greater interest in fairness, equality, and loyalty. “The organization provides benefits to its employees beyond its legal and financial obligations (voluntary) and employees feel obliged to pay back these voluntary investments. Consequently, it stimulates direct social exchange relationships between employees and their organization” (16, p. 567). There is evidence that higher levels of ICSR components are positively associated with higher organizational commitment. For example, Tarcan et al. (22) proposed that organizational commitment reacts positively to initiatives related to transformational leadership, job satisfaction and organizational trust. In addition, in a study of 191 hotel employees in Indonesia (23), a confirmatory factor analysis revealed that ICSR practices were associated with a decrease in employee turnover and increase in organizational commitment (β = 0.27). These findings suggest that ICSR practices may help employees to address challenges and opportunities from the external environment and facilitate the development of a highly talented, motivated, and committed workforce (24). Mory et al. (7) also reported a positive and significant relationship between ICSR and both affective and normative components of organizational commitment. 

The relation between CSR and organizational commitment have been explored, as “more socially responsible corporations are more attractive to potential employees and that they may therefore benefit from larger applicant pools […] and a more committed workforce because ‘employees will be proud to identify with work organizations that have favourable reputations’ (25, p. 1702). 

Farooq et al. (2014) reported that CSR toward employees (which includes actions as career opportunities, organizational justice, family-friendly policies, safety, job security, and union relations) is the strongest predictor of employees´ trust, identification, and affective organizational trust (16). 

In relation to ICSR, Brammer et al. (2007) reported a positive relation between organizational justice and organizational commitment, in particular, these authors understand organizational justice as an special kind of ICSR, as “beneficial actions directed at employees create a reason for employees to reciprocate with their attitudes and their behaviours. At the same time a positive relationship may be expected between procedural justice and affective commitment because employees may be expected to identify with ethical organizations” (25, p. 1705). Following this argument, we can expect that, as all the dimensions considered in the ICSR Q are beneficial actions directed to employees, there could be a relation between these seven actions and organizational commitment.

Affective commitment refers to identification, involvement, and emotional attachment to an organization. Therefore, employees with strong Affective commitment stay with the organization because they want to. Normative commitment refers to commitments based on a sense of obligation to the organization (26).

Recent studies based on theory of social exchange establish a positive link between employee’s perceptions of their companies and CSR, developing CSR practices creates a good reputation among employees. Likewise the CSR toward employees the firm´s actions must ensure the well – being and support of its employees, including career opportunities, organizational justice, family – friendly policies, safety, job security, and union relations (16). 

A study conducted in different business sectors in Romania investigated the relationship between factors such as organizational commitment and social responsibility towards employees, and the survey response rate was 70%, showing a correlation of 0.447. The results showed that social responsibility to employees was positively correlated with organizational commitment and was the variable with the highest value, probably because the sample was composed of employees. This was one of the first articles to show the construction of a solid relationship between employees and employers (27).

We therefore expect ICSR to be positively and significantly related to affective and normative organizational commitment.

Hypothesis 2: ICSR is positively correlated to affective organizational commitment (hypothesis 2a) and normative organizational commitment (hypothesis 2b).

Workers with permanent contracts tend to have more advantages than workers with fixed-term contracts. Employees with permanent contracts react to job stability with increased well-being and organizational commitment (28), while employees with fixed-term contracts tend to exhibit low satisfaction and limited organizational support, and as a result commitment to the organization decreases and acquired knowledge is lost when employees leave (29). 

Numerous studies have revealed that unemployment has negative psychological and physical effects on the unemployed. Social dynamics in employment have a positive impact on workers' self-awareness and well-being. However, the employment relationship has changed considerably and, with the liberalization of labor laws in the mid-1980s, the number of fixed-term contracts has increased. In general, fixed-term employment has increased in most OECD countries over the last 20 years, and in Spain workers with fixed-term contracts seem to be trapped in a situation of fixed-term contracts and unemployment (30).

Workers on fixed-term contracts are more likely to lose their jobs than those on permanent contracts, simply because their contracts end for a short period of time, usually one to two years. The economic hardships faced by the unemployed, in addition to the potential functions that employment provides and those that are important to personal well-being, are the consequences of the inability of fixed-term workers to plan and control their own lives, given the nature of employment as they leave their jobs for short periods of time. Job insecurity due to the risk of unemployment and relatively low productivity in some jobs with long-term contracts, in addition to being associated in most cases with relatively low wages and poor working conditions, so that job insecurity can affect the mental health of workers, worsening their health status (30).

There are several reasons why employers hire workers on fixed-term contracts, companies turn to flexible staffing to reduce contract labor and associated costs. By using flexible staffing arrangements, companies can adjust staffing levels based on fluctuations in workload during a day, week, month or year, avoiding the need to maintain staffing levels continuously during peak hours. Similarly, companies can hire temporary agents when employees are sick or on vacation, rather than paying overtime to regular workers when employees are absent. Another reason is that companies are responding to competitive pressures to reduce labor costs, and managers are looking to make their organizations more efficient, in part, by using more flexible contracts (31).

Taking into account that employees under permanent contracts are expected to have higher employment stability, and are compensated with higher levels of practices of ICSR, we will use the employment contract as another way to validate the ICSR Q, as if our validation is able to detect differences between type of contract. Thus, employees with permanent contracts are likely to have higher levels of ICSR practices. We thus expect that workers with permanent contracts report higher values of ICSR than workers with fixed-term contracts.

Hypothesis 3: Workers on permanent contracts report higher evaluations of ICSR than workers on fixed-term contracts.

Point # 3: Method

Misunderstanding with the information included. In the measures, it is mention that both English and Spanish versions of questionnaires are included and only the Spanish version is presented. 

A. Thank you for your observation, we have included both English and Spanish versions (pp. 33-37)

The internal corporate social responsibility is measured by 48 items, however in the Spanish version of questionnaire only 45 questions are included. This requires a further explanation.

A. Thank you for your observation, as we have explained in p. 19, we eliminate three questions due to a low loading of the items in their dimension, this is a procedure recommended in the validation of questionnaires (Hair et al., 2010). In addition, we explain that the elimination threshold was .50. Now this section reads as follows (p. 19):

We analyzed 48 items of the ICSR Q to test whether the Spanish - speaking version retained the structure reported in the original article. After initial analyses, three items were dropped due to small loadings of the items within the construct: one item from skill development, one from work-life balance, and one from tangible employee involvement. The loadings were below .50 which is below the recommended threshold (41).

Hair JF, Black WC, Babin BJ, Anderson RE. Multivariate Data Analysis (7th Edition). 7th ed. Upper Saddle River, NJ, USA: Prentice Hall; 2010.

Could the data collection bias the results? Almost 50% of the sample comes from a university and the rest from private participants in a human resource management course.

A. Thank you for your observation, it is important to note that the original ICSR Q was developed using the sample of a single pharmaceutical organization. We consider that our sample is heterogeneous enough to include a wide range of job positions and organizations, as the sample included jobs from 20 of the 21 economic activities listed in the International Standard Industrial Classification of All Economic Activities (ISIC) (p. 16). In addition, we included an specification of the original German sample in p. 9-10:

The original questionnaire was developed using a sample of 2081 employees from an international operating pharmaceutical company with its headquarters located in Germany, the sample distribution was 50.6 male, 17.9% were under 29 years, 24.8 from 30-39 years, 35.2% from 40-49 years, and 22% were older than 50 years. The tenure of the sample was up to 2 years 17.3%, from 2-5 years 16.7%, from 6-10 years 22%, and 10 years and more 44% (7).

Much more description is needed to improve confidence in the data being used, for instance, when the data was accessed?

A. Thank you for your observation. We consider that the procedure we used in the collection of our sample was quite rigorous and we have to highlight that before we collected the data, the ethics review board of the Business department at Pontificia Universidad Javeriana approved the procedure before the study began. In addition, we included in p. 19 the information in which the data was collected (2019)

Point # 4: Results

The comments in this section are very superficial. And additional explanation and argumentation of the results is needed. 

A. Thank you for your observation. Maybe the reviewer refers to the discussion section, as we consider that the results section is very descriptive and precise (pp. 18-23). We have expanded the discussion and conclusion sections. Now these sections reads as follows:

Discussion

Our aim was to validate the Spanish - speaking version of the ICSR Q proposed by Mory et al. (7). Confirmatory factor analysis indicated that the seven original factors were preserved due to adequate levels of reliability and validity. Like the original instrument, the Spanish - speaking version showed a significant relation with both affective and normative organizational commitment. Finally, as expected, workers with permanent contracts also reported higher levels of ICSR and organizational commitment. With these results, we have accomplished our five original objectives: 1) to present an Spanish version of ICSR Q, 2) to confirm the seven-dimension structure of the scale, 3) to determine the internal consistency of the ICSR Q, 4) to explore the relation between the ICSR Q and organizational commitment, and 5) to examine the association between ICSR dimensions and type of contract.

By validating the Spanish - speaking version of the ICSR Q, this study contributes to the field of internal social responsibility by confirming the relevance of the identified factors despite differences in cultural context, workforce, economic sectors, and hierarchical levels. The fact that Colombian workers in our sample evaluated questionnaire items similarly to German workers is not surprising, since over the last 25 years Colombia has undergone increasing integration into world markets through attraction of multinational companies as well as adoption of international practices of human skills management. Furthermore, uncertainties felt by workers as the economy and labor relations become more flexible were also expressed.

Our results are in accordance with previous efforts as 26001, KLD, or OSHAS 18001, as all these standards and indexes have components that look to assess the level of responsibility that organizations have with their own employees in components as work conditions, safety, training, or diversity. However, we consider that the Spanish-speaking version of the ICSR Q offers additional advantage to these standards and indexes as are the organizations which can apply the questionnaire and obtain valuable information about how the organization is perceived among their employees, offering and insightful evaluation of each one of the seven dimensions of ICSR, either for internal change or as first approach to ICSR before opting for a more complex evaluation of ISO or OSHAS.

At the theoretical ground, we have tested an instrument based on the social exchange theory and our results are in accordance with previous research highlighting the exchange between “good employee practices” and the retribution that employees offer to their organizations trough organizational commitment, not just as normative commitment, in which commitments is based on a sense of obligation to the organization, but also as affective commitment, which is based on the identification, involvement, and emotional attachment to an organization. These positive relations lead to higher job satisfaction, and lower turnover rates and absenteeism.

We argue that the inclusion of a greater number of items per dimension in the Spanish - speaking version was beneficial, since higher factor loadings were achieved than in the original study. The only exception was the empowerment component, which showed adequate but slightly inferior indicators than the original instrument. 

In relation to contracts, our results have offered additional evidence of the importance of a long-term relation between the organization and its employees, as they perceived the relation with the organization in a more deep level (i.e., more responsible), and then, they feel that they are really important to the organization, and in return, they offer additional organizational commitment. In the comparison between the groups with permanent and fixed-term contracts, empowerment was the only factor that showed no difference among groups; this may be explained by cultural traditions in Colombia. While the empowerment component reflects the autonomy of workers to perform their tasks, Colombia has been traditionally classified as a collectivist country with a high distance to power (19). Such values may clash with the promotion of individuality typical of Anglo-Saxon or Germanic countries. The explanation may be that in contrast to the other ICSR Q variables, it does not relate to a skill management practice, but rather reflects a set of values cutting across skill management practices. As for the remaining dimensions, the greatest differences were found in job stability (directly related to type of contract), and in affective organizational commitment, which may be related to other job outcomes, such as job performance or intention to stay.

Social exchange theory postulates that voluntary benefits from employer to employee, among other actions promote employee well-being (16). Employees in our sample positively evaluated organization investments in well-being and affective commitment, with our findings suggesting that ICSR also positively influences their attitudes. The results are thus consistent with the theory of social exchange and with initiatives promoting the development of employee skills.

Implications for practice and research

Our results have several implications for professional practice and research. Among the practical implications, the availability in Ibero-American countries of a reliable and valid practice measurement tool may allow the evaluation of ICSR practices by employees. Monitoring may serve a dual purpose: to evaluate organization ICSR or skill development (diversity management, work-life balance), and to use the relationship between the questionnaire and organizational commitment as an indirect indicator of the intention of permanence and involvement, especially with the strategic organization personnel.

Implications for research follow from the validation of internal social responsibility. Most research so far has traditionally focused on external stakeholders to the organization, and as far as employees are concerned, research has focused only on compliance with labour laws. Therefore, the validation of a measurement instrument in Spanish - speaking may help to consolidate the ICSR phenomenon in Ibero-American contexts.

Finally, we hope that the validation of this questionnaire will promote research on ICSR in Ibero-American countries, targeting the identification of ICRS profiles in different countries of the region, or the relationship of ICRS dimensions to other outcome variables such as organizational performance or talent keeping.

Limitations

Despite our positive results, two limitations can be highlighted. The first is the sampling method. Although two different samples were obtained, they were collected in the urban context of Bogotá, where there is a greater representation of certain service organizations and the quality of job positions is higher; however, the proportion of the informal labor sector is around 50%. Although our conclusions characterize the sample of urban workers, future research should aim to represent other regions and economic sectors.

Non-representative samples may not adequately describe the workforce as a whole, and estimates may include a selection bias preventing generalization of results to the Colombian labour context. For this reason, an alternative longitudinal analysis may be able to investigate the effects of ISCR factors on affective and normative commitment from a different perspective. Future research should also focus on companies representing different environments or economic activities.

Conclusion

Our validated Spanish - speaking version of the ICSR Q proposed by Mory et al. (7) offers a reliable instrument to identify factors constituting ICSR and its relationship with organizational commitment. Our investigation provided evidence of preservation in the Colombian labour context of the same seven factors underlying the original ICSR Q. 

The ICSR Q also offered evidence of the link between internal responsible practices for employees and organizational commitment, a relation that have also been reported in general CSR. Finally, our findings also give additional evidence about the importance of long term-contracts in the organizational commitment of employees, at the normative and affective level.

We expect our study to stimulate further academic research as well as further ICRS practices in the business context.

---

## [Decision Letter · Decision Letter 1]

11 Mar 2022

PONE-D-21-30685R1Spanish-speaking validation of the internal corporate social responsibility questionnairePLOS ONE

Dear Dr. Bayona,

Thank you for submitting your manuscript to PLOS ONE. After careful consideration, we feel that it has merit but does not fully meet PLOS ONE’s publication criteria as it currently stands. Therefore, we invite you to submit a revised version of the manuscript that addresses the points raised during the review process.

Both reviewers and myself are very satisfied with the thorough revision that you made. However, one slight problem subsists regarding the availability of the data used to do the validation. You declare that they are available without restriction in the text or auxiliary information, but we are not able to locate them. You should include a link to a public repository or provide them in appendix so that the paper can adhere to the editorial guidelines of the journal.

We look forward to receiving your revised manuscript.

Kind regards,

Iván Barreda-Tarrazona, PhD

Academic Editor

PLOS ONE

Journal Requirements:

Reviewers' comments:

Reviewer's Responses to Questions

**Comments to the Author**

1. If the authors have adequately addressed your comments raised in a previous round of review and you feel that this manuscript is now acceptable for publication, you may indicate that here to bypass the “Comments to the Author” section, enter your conflict of interest statement in the “Confidential to Editor” section, and submit your "Accept" recommendation.

Reviewer #1: All comments have been addressed

Reviewer #2: All comments have been addressed

2. Is the manuscript technically sound, and do the data support the conclusions?

Reviewer #1: Yes

Reviewer #2: Yes

3. Has the statistical analysis been performed appropriately and rigorously? 

Reviewer #1: Yes

Reviewer #2: I Don't Know

4. Have the authors made all data underlying the findings in their manuscript fully available?

Reviewer #1: Yes

Reviewer #2: No

5. Is the manuscript presented in an intelligible fashion and written in standard English?

Reviewer #1: Yes

Reviewer #2: Yes

6. Review Comments to the Author

Reviewer #1: The authors have carried out all the suggestions so the paper is now ready to be published in this journal.

Reviewer #2: All my previous comments have been addressed.

Please note that following PLOS Data policy, the authors should make all data underlying the findings described in the manuscript fully available.

7. PLOS authors have the option to publish the peer review history of their article (what does this mean?). If published, this will include your full peer review and any attached files.

Reviewer #1: No

Reviewer #2: No

---

## [Author Response · Author response to Decision Letter 1]

18 Mar 2022

All modifications are highlighted in the document 

Response to reviewer # 1

Point # 1: The authors have carried out all the suggestions so the paper is now ready to be published in this journal.

Thank you for all your previous comments, with your help, our manuscript is now clearer and more focused.

Response to reviewer # 2

Point # 1: All my previous comments have been addressed. Please note that following PLOS Data policy, the authors should make all data underlying the findings described in the manuscript fully available.

Thank you for all your previous comments, with your help, our manuscript is now clearer and more focused.

We have included a “supporting information section” that will link the full data set. This section now reads as follow (p. 27):

Supporting information

S1 Dataset. Study sample (.SAV)

In addition, we uploaded the database to the PLOS ONE submission center (we forgot to upload the database during R1)

---

## [Editor Report · Decision Letter 2]

28 Mar 2022

Spanish-speaking validation of the internal corporate social responsibility questionnaire

PONE-D-21-30685R2

Dear Dr. Bayona,

We’re pleased to inform you that your manuscript has been judged scientifically suitable for publication and will be formally accepted for publication once it meets all outstanding technical requirements.

Kind regards,

Iván Barreda-Tarrazona, PhD

Academic Editor

PLOS ONE
---

## [Editor Report · Acceptance letter]

6 Apr 2022

PONE-D-21-30685R2 

Spanish–speaking validation of the internal corporate social responsibility questionnaire 

Dear Dr. Bayona:

I'm pleased to inform you that your manuscript has been deemed suitable for publication in PLOS ONE. Congratulations! Your manuscript is now with our production department. 

Kind regards, 

on behalf of

Dr. Iván Barreda-Tarrazona 

Academic Editor

PLOS ONE